# GOING GREEN: The Effectiveness of a 40-Day Green Exercise Intervention for Insufficiently Active Adults

**DOI:** 10.3390/sports7060142

**Published:** 2019-06-13

**Authors:** Nicholas Glover, Scott Polley

**Affiliations:** Alliance for Research in Exercise, Nutrition and Activity (ARENA), School of Health Sciences, University of South Australia, Adelaide 5001, Australia; scott.polley@unisa.edu.au

**Keywords:** green exercise, adherence, compliance, health, outdoor and adventure activities

## Abstract

Increasing physical activity and reducing sedentary behavior is an economic and health priority. This Green Exercise (GEx) study reports on a 40-day physical activity intervention to increase physical activity that primarily used outdoor recreation activities. Adherence, compliance, blood pressure (BP), total cholesterol, anthropometry, strength, dynamic stability, and cardiovascular fitness were assessed 1 week prior and immediately following the 40-day intervention. The results then were compared with a larger study that used the same methodologies but for the exception of primarily indoor physical activities. Results from this study showed similar improvements in health measures to the comparative indoor-based physical activity program with increased adherence and compliance. Improvements in wellbeing were also noted. This GEx study suggests that exercise programs that seek to increase physical activity levels of insufficiently active adults may benefit from including outdoor recreation activities within the program and may also increase participant mental health and general well-being.

## 1. Introduction

Western societies are much less physically active than our pre-industrial forebears and far less active than the humans who were our ancient ancestors [1]. Technological advances have resulted in changes to the way we live following the onset of the agricultural and industrial revolutions and the digital age [2]. These changes now mean that we are less likely to be engaging in physically active behaviors as we spend more time in built environments and less time in agrarian or natural environments.

Physical Inactivity is thought to be the fourth leading risk factor for global mortality (following high blood pressure, smoking and diabetes) and is the main cause of 21–25% of breast and colon cancers, 27% of diabetes and 30% of ischemic heart disease [3]. Physical inactivity, along with unhealthy diet, tobacco use, and excess alcohol consumption is a key contributor to 28 million deaths from non-communicable diseases [3]. Increasing physical activity (PA), eating well and reducing smoking is thought to have the potential to prevent 80% of premature heart disease, 80% of type 2 diabetes and 40% of cancers [3]. Physical activity is also known to positively affect mental health [4] with depression now the leading cause of disability world-wide, affecting an estimated 350 million people [5]. Despite positive benefits of PA, estimates from Australian state and territory surveys suggest that less than 50% adults are achieving sufficient levels for health [6].

Addressing sedentary behavior is now a global health challenge [7]. Direct PA interventions have been shown to positively affect behavioral change in terms of participants’ adherence and compliance to exercise following a program [8,9]. Recent systematic reviews provide some direction for PA interventions to be more successful in promoting adherence (the extent to which participants continue in a program to its conclusion) and compliance (generally defined as the extent to which participants meet a prescription of PA) [8,9,10]. Intervention strategies that engage participants in PA behaviors were more likely to be effective than cognitive variants [11]. Group-based and educational interventions were found to be more effective in the short-term when compared to home-based strategies [12]. Long-term interventions were found to be more effective for older (compared to middle-aged) populations and when using booster strategies such as providing educational materials [9]. The use of monitoring devices [13] shows promise for increasing adherence and compliance for those with identified conditions [13,14].

Barton and Pretty [15] theorized that PA in outdoor and natural environments provides increased benefits compared with exercise in indoor environments. A recent term, ‘Green Exercise’ (GEx) was coined and defined somewhat broadly as exercise in the presence of nature [15]. Supporting this theory, a small number of studies comparing PA in ‘non-green’ and ‘green’ spaces suggest that natural views and natural environments may increase PA participation [16], intensity levels [17,18], physical health [10,18,19,20,21,22,23,24,25,26,27], and mental health benefits [10,17,18,25,26,27,28,29,30,31,32,33,34,35].

No studies were found prior to this intervention that assessed comparative adherence or compliance rates as part of intervention programs for insufficiently active adults; therefore, any beneficial effect GEx might have on increasing levels of PA has not previously been tested in this population.

Withdrawal from PA interventions and general exercise programs is a recognized problem [36,37,38,39]. It is theorized that adherence is affected in part by the environment in which people exercise [10,33] and that GEx might positively influence adherence [30,40,41,42]. Nearly half of indoor sports participants drop out within the initial 6 months, whereas it has been reported that walking outdoors is a preferred form of exercise to maintain adherence [43].

Theories about possible increases in exercise adherence and compliance with GEx include attention distraction [30], costs associated with exercise [33] and the biophilia hypothesis proposed by Kellert and Wilson [44] suggesting humans have an innate attraction to nature. Mackay and Neill [32] theorized that the greater the ‘greenness’ of an environment, the greater the potential benefit to those immersed in it.

To investigate the comparative role of GEx with other exercise interventions in promoting adherence and compliance, a previously successful larger scale 40-day PA intervention by Norton et al. [45] that primarily used indoor environments [45] was replicated in structure, with the exception of the group activities that replaced indoor-based with outdoor-based pursuits. Norton and colleagues’ [45] study incorporated three intervention arms: (1) a pedometer-based group with no direct facilitation (*n* = 251); (2) an active control group consisting of sufficiently active subjects continuing to meet recommended weekly requirements for PA (>150 min/week; *n* = 135); and (3) an instructor-led cohort utilizing group-based and individual exercise sessions, largely indoors (*n* = 148). Results from Norton and colleagues’ [45] study indicated that this latter intervention arm was most successful at improving PA participation and associated health measures, and therefore, was modelled for this study, but with a GEx focus.

The primary aims of this study were to measure program adherence and exercise compliance among participants undertaking a 40-day daily PA intervention based on GEx. Secondary aims included to determine the changes in a range of physical, physiological and psychological variables following the intervention and to compare the changes in the GEx intervention with those previously reported by Norton et al. [45]

## 2. Materials and Methods

Recruitment for the 40-day GEx Intervention occurred through numerous mechanisms: Firstly, via email throughout the University of South Australia and a number of South Australian government departments; secondly, through a news story in a local newspaper; and thirdly, via recruitment posters placed around the University City East campus.

Inclusion criteria for both studies was insufficiently active (<150 min/week of moderate-to-vigorous PA assessed using the Active Australia Survey [46]); otherwise healthy; 18–60 years of age; available for a 40-day PA program. The structure included three instructor-led group sessions per week, being 19 group sessions in total over the 40 days. Group sessions ran for a minimum of 30 min and were planned to progressively increase participants’ energy expenditure (EE) requirement each session. On non-group days, participants were asked to undertake their own exercise session for a minimum of 30 min, totaling 21 of the 40 sessions. The program ran throughout April, being autumn in the southern hemisphere. Norton’s [45] study had 11 indoor sessions undertaking training activities that included circuit training with weights, stair climbing, stretching and resistance activities, aerobics, and spin-cycling classes. Six activities took place at nearby city parks (jogging, soccer, stretching), with two group sessions planned to take place in more natural environments. The program ran during autumn, winter and spring. By comparison, this program of GEx (Table 1) included only outdoor recreation activities in local, easily accessible green spaces, using the criteria outlined by Mackay and Neill [32].

The settings included places such as parklands, riverside settings, conservation parks, and marine and coastal environments (Figure 1). Activities included walking, low organization team games, challenge activities, yoga, kayaking, cycling, rock-climbing, and orienteering all conducted in an outdoor environment. The program itinerary used freely available and conveniently located (near-city) public green spaces with the intention to introduce participants to a diverse range of recreational activities that could be undertaken beyond the program (Table 1).

Participants attended the group sessions three times per week (Tuesdays, Thursdays and Sundays) for activities conducted by trained instructors and undertook an activity of their own choice on alternate days (Table 3). The activity sessions were designed to expend approximately 800 kJ in the first week increasing by approximately 200 kJ in each subsequent week. All sessions included a 10-min warm-up and cool-down with a stretching period. Weekday sessions lasted 60 min and Sunday sessions around 90 min. Where possible, the core of the session had subjects working between 60–80% of age-predicted HR_max_ (220-age in years). This was not always attainable due to the nature of the activities, for example rock climbing which requires bouts of intense activity interspersed by rest.

Using Norton’s [45] study as the baseline, a sample size of 19 was required to detect changes at α = 0.5 and power = 0.8. Although 23 participants commenced the program, only 17 achieved full participation with pre- and post-intervention testing.

As with Norton’s [45] study, participants were tested 1 week immediately pre- and 1 week post-study using the same protocols for a range of physical health variables.

Psychological variables were also assessed for this GEx intervention, although not in the Norton [45] study.

The major variables assessed included: blood pressure (BP), measured according to the technique recommended by the American Heart Association [47]; height was measured with the subject in light clothing and bare feet using the stretch stature method [48]; weight where subjects were weighed in minimal clothing, following an 8 hour fast and after voiding; body mass index (BMI) was then derived from the height and weight measures; girth was taken at the level of the narrowest point between the lower rib and the iliac crest when viewed from the front; hip girth was taken at the level of the greatest posterior protuberance of the buttocks; the waist–hip ratio (WHR) of subjects was determined by dividing the waist girth by the hip girth; grip strength using an isometric dynamometer (Takei Kiki, Tokyo, Japan); total cholesterol was measured using finger-tip blood samples from 8-hour fasted patients; aerobic fitness (mL·kg^−1^·min^−1^) was predicted using a non-gas analyzed sub-maximal test conducted on an electroncally braked cycle ergometer (Ergoselect 200). The average heart rate (HR) in the final 15 s of each workload was used to construct a regression line for each person. The regression line was extrapolated mathematically to their age-predicted maximal HR (HR_max_). On this basis, an estimate was made of the power output (W_max_) they would have achieved at HR_max_, and the corresponding oxygen uptake was calculated using: VO2_max_ (mL·kg^−1^·min^−1^) = ([W_max_/9.81] × 60 × 2 + [3.5 × Weight])/Weight. Prior to testing, the validity and reliability of tests were assessed using 5–7 repeated tests on the same subject (Table A1, Appendix A).

To assess that sufficient levels of PA were achieved, Polar brand HR monitor watches were used, supplemented by self-reported ratings of perceived exertion [49] and activity diaries. Participants were instructed to program measured VO2_max_ and HR_max_ values into the Polar S610 watch [50]. The watch uses this data and with its proprietary software estimates EE, accounting for subject gender. Crouter and colleagues [51] found that using actual measured values for VO_2max_ and HR_max_ resulted in a 4% error (SD ± 10%) in EE.

Instructors provided leadership, instruction, feedback, and guidance during the critical early phase of the activities where participants are more likely to drop out [52]. Many of the outdoor and recreational activities were such that participants were undertaking them for the first time or had not undertaken them since childhood.

The psychological assessment for wellbeing was measured using the self-administered questionnaire: Personal Wellbeing Index—Adult (PWI-A [53]). Self-efficacy was measured using the questionnaire: The Physical Exercise Self-Efficacy Scale [54]. Participants’ depression, anxiety and stress were measured using the DASS21 questionnaire [55].

Participants in this study met the criteria for classification as insufficiently active (PA level < 150 min/week) by completing one Active Australia Survey [46], a 7-day recall questionnaire. It is recommended by the Department of Health [56] that adults ‘accumulate 150 to 300 min (2 ½ to 5 h) of moderate intensity PA or 75 to 150 min (1 ¼ to 2 ½ h) of vigorous intensity PA, or an equivalent combination of both moderate and vigorous activities, each week.’ National and state-level surveys have consistently found that approximately half of all adults in Australia do not meet the minimum guidelines [57].

Participants’ pre-intervention PA level averaged 84 min/week (range 0–148 min/week). This placed them in a risk factor category for low PA patterns being, on average, in about the lowest third of PA levels among adult South Australians [58]. Participants were mostly aged in their 40s or 50s (48.3 ± 10.2 years) and had poor cardiorespiratory fitness (mean ± SD, VO_2max_ = 25.4 ± 10.6 mL/kg/min), with many showing other risk factors such as hypertension (29%) and high cholesterol (47%; including those on prescription cholesterol-reducing medication).

Average BMI for participants was 30.2 kg·m^−2^ pre-intervention (range 23.1–46.2 kg·m^−2^). Low levels of PA and high body fatness levels significantly increase the risk for chronic conditions such as diabetes and metabolic syndrome, and developing coronary heart disease [59].

Descriptive information was calculated for all variables measured. Pre- and post-comparisons within the GEx sample group were made using paired *t*-tests, and those reaching significance (*p* < 0.05) were reported. The original dataset (*n* = 622) for Norton’s [45] intervention was used in the analysis of the significance of the pre-post changes in the current cohort. Comparisons with those results were made using repeated measures analysis of variance (ANOVA). Chi squared analysis was used to compare rates of adherence and compliance within and between interventions.

Ethics approval for this project (Ethics Protocol P017-06) was gained from the University of South Australia Human Research Ethics Committee.

## 3. Results

### 3.1. Participants

Participant pre-intervention data for those that completed the program are shown in Table 2. Mean ages of participants were 48.8 years for males and 47.8 years for females; the youngest and oldest within both groups being 28 and 59 years respectively. The numbers of males (*n* = 8) and females (*n* = 9) in the finishing group were relatively even.

### 3.2. Adherence and Compliance

Inquiries were fielded from 197 members of the public with the offer of either a group-focused (a concurrent study not reported here) or outdoor-focused exercise program. Twenty-six screened participants were assigned the outdoor-focused exercise group, with the first exercise session commencing with 22 participants, of which 17 participants (77% adherence) completed the program and returned for post-intervention testing. Withdrawals were due to reported unrelated medical issues, family circumstances and employment commitments.

Data collected from the Polar Heart Rate Monitors (HRMs) were used to assess daily compliance rates, confirmed by PA Diaries and group session attendance records. Of a possible total of 680 participant-days, there were 397 (58%) recorded on the HRMs, being the days on which participants complied with the requirements of the intervention (≥30 min/day of recorded PA). This is a conservative calculation because attendance records showed numerous instances where participants attended the group sessions but either forgot to record the session on their HRM or had technical problems and no recordings were present when downloaded. Using individuals’ PA diary records as well as HRM and attendance data resulted in a final compliance rate of 74%.

There was a gradual decrease in compliance for both group and individual days across the first 3 weeks, and compliance was lowest in week 5 (group 59%; individual 29%). The mean rate of compliance on group days was 77%, which was higher than on individual days (46%). Chi squared analysis determined that compliance on group exercise days was higher than expected, but lower than expected on individual exercise days. The difference in compliance between group and individual exercise days was significant (*p* < 0.0001). 

Using a second measure of compliance, it was found that of the participants who completed the program, there were 16 (94%) who achieved sufficient levels of PA (≥150 min/week) at post-testing.

### 3.3. Physical Activity

Figure 2 shows the daily recorded mean values for exercise heart rate (HR) and estimated energy expenditure (EE) matched to the corresponding group session or day of individual exercise. Mean HR values ranged from 102 to 138 on individual exercise days and from 103 to 134 on group exercise days. The mean energy expenditure (EE) on individual exercise days was 1076 kJ and ranged from 707 kJ to 1531 kJ. On group exercise days, the mean EE was 1539 kJ and ranged from 1088 kJ to 2470 kJ. Values for each session are shown in Table 3.

### 3.4. Changes to Physical and Physiological Health Following the Intervention

Changes to values for health and well-being are shown in Table 4. Small (but not significant) absolute decreases were found for weight, BMI and waist. 

The change for hip reached statistical significance (*p* = 0.036). Further significant changes were seen for total cholesterol (*p* = 0.026), aerobic fitness (*p* = 0.002), dynamic stability (*p* = 0.038) and all categories of PA minutes (*p* < 0.001). No adverse changes to variables of any category were observed.

Results of the outdoor PA intervention were compared to those of the Norton [45] study using repeat-measures ANOVA to check for any significant intervention x time interaction differences in a range of variables. There were no significant differences between the intervention changes in all but two categories, meaning that the GEx intervention resulted in improvements of a similar nature to those seen following the indoor-based intervention for almost all variables (except grip strength and vigorous PA minutes).

### 3.5. Changes to Mental Health and Well-Being Following the Intervention

Although not investigated in Norton’s [45] study, of interest for the GEx study was the potential for changes in participant mental health and well-being. This additional investigation was conducted using Personal Wellbeing Index—Adult [53], The Physical Exercise Self-Efficacy Scale [54] and the DASS 21 questionnaire [60]. The GEx intervention enhanced outcomes for four of the five psychological variables, with significantly improved mean scores for well-being (*p* < 0.001), depression (*p* < 0.001), anxiety (*p* = 0.042) and stress (*p* = 0.004). Raw scores for self-efficacy also increased but not to statistical significance (Table 4).

Figure 3a plots the mean changes for well-being and self-efficacy from pre- to post-intervention. Improvements are represented by increased scores. Figure 3b plots the mean changes for depression, anxiety and stress from pre- to post-intervention. Improvements are represented by decreased scores.

Although well-being increased significantly (*p* < 0.001) across the outdoor PA intervention (Table 4), no significant intervention x time relationship was detected. Significant relationships were detected between the starting value and the change in value for self-efficacy (*p* < 0.001), depression (*p* < 0.001), anxiety (*p* = 0.007), and stress (*p* = 0.003). This effectively means that the lower a starting score for self-efficacy (or the higher a starting score for depression, anxiety or stress), the greater the likelihood a positive change will occur.

## 4. Discussion

This intervention study sought primarily to measure adherence and compliance to a GEx-based program of PA. Secondary aims were to improve the health and well-being of participants and to compare the extent of change against a program that utilized primarily traditional, indoor-based physical activities.

Physical activity interventions may only be successful if participants comply with protocols and adhere to a program. Encouragingly, this GEx intervention recorded similar (77%) adherence when compared with the indoor program (84%), suggesting the potential for strong participant retention with PA programs in green spaces. Adherence is likely to vary with climate and other setting-specific factors; for example, warmer weather and longer daylight hours in an aesthetically pleasing setting may strengthen participation and should be considered when setting a program.

Compliance was also comparable between the GEx (58.1%) and indoor (62.6%) programs. Findings from both interventions suggested future PA interventions might benefit from including more group-based sessions where higher compliance was recorded, compared with individual sessions of exercise. For this study, weekly compliance (≥150 min/week) could be reached by attending the three group sessions only, which may have acted to demotivate participation in individual sessions where compliance was much lower, for example week five (29%).

Further results indicate that similar outcomes (10 of 12) were achieved for the physical and physiological measures. This result would indicate that beneficial PA can be achieved without the need for costly, tailored indoor spaces and equipment, as the majority of the GEx program was conducted in public green spaces, with little or no equipment. Further benefits to participants were reported in the form of psychological measures, all showing an improvement pre- to post-intervention, four of five being significant. Although these results did not have a direct comparator, they would appear to support the growing number of studies [10,15,17,18,32,61,62,63,64,65] showing the potential for GEx programs to improve the health and well-being of participants across a range of measures.

Limitations to this study must be acknowledged, particularly related to sample size. The number of participants was modest (and not to the statistical power calculation) where a larger sample would increase the confidence that the results reflected potential changes in the broader population. Moreover, the mean age of these participants (48 years) was much higher than for the comparative group (35 years). This age difference is likely to influence participants in many ways, such as time availability, motivation, physical and mental condition, and other life circumstances. Additionally, a control group would have improved study design and allowed for direct comparison for the assessments undertaken. A final limitation to acknowledge is that disparate compliance rates (between individuals, or by individuals from week-to-week or in group versus individual sessions) are likely to have resulted in varied impacts on the health and well-being outcomes recorded. Greater consistency in compliance rates among individuals and by individual participants across the program would allow for more confident conclusions to be drawn on the effectiveness of GEx to improve health and well-being.

An informal follow-up at 12 months provided a lot of anecdotal evidence to indicate that some participants had continued to be active in small groups, for example with “weekly outdoor exercise excursions” (email correspondence, 8 April 2013). Participants also reportedly continued to receive the GEx “benefit… that is both physical and emotional/mental” (email correspondence, 8 April 2013). 

## 5. Conclusions

In conclusion, with the considerable limitations in mind, this study would appear to support GEx as a viable alternative to other programs by offering the potential for similar health and well-being results when compared with indoor exercise programs. Further, for those seeking psychological benefits from exercise, GEx has provided positive outcomes for almost all participants of this study.

For some, GEx may be a preferred form of activity, particularly for those who have an aversion to joining gyms or clubs, have financial constraints, or have issues with accessing traditional facilities. Green spaces are generally free to use and prevalent in developed cities; however, this is not always the case. A lack of access or other factors such as a real or perceived lack of safety may be a deterrent to participation. A focus by government on creating and maintaining natural outdoor spaces may provide the impetus for people to engage in GEx, a low-cost and effective means of improving physiological and psychological health and well-being when compared with indoor exercise requiring specific facilities and equipment.

It is recommended that further research into GEx be undertaken, particularly to follow up its potential to enhance mental health and well-being and the associated effects on adherence and compliance to PA programs. 

## Figures and Tables

**Figure 1 sports-07-00142-f001:**
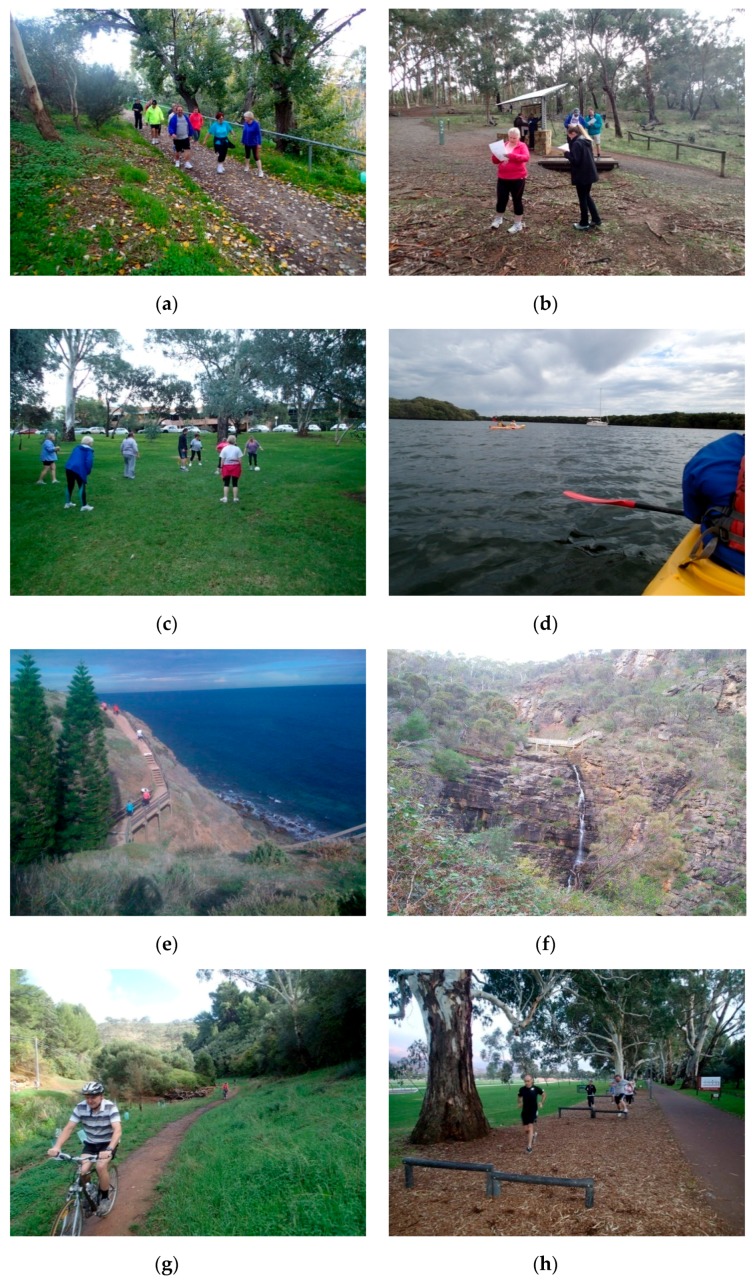
Samples of group exercise sessions and locations around Adelaide, SA. (**a**) Walking along River Torrens Linear Park; (**b**) orienteering in Belair National Park; (**c**) team games at Victoria Park; (**d**) kayaking at the Adelaide Dolphin Sanctuary, Port River; (**e**) walking the Marion Coastal Walking Trail, Marino; (**f**) rock climbing in Morialta Conservation Park; (**g**) cycling in Brownhill Creek Recreation Park; and (**h**) sweat-track workouts at Victoria Park.

**Figure 2 sports-07-00142-f002:**
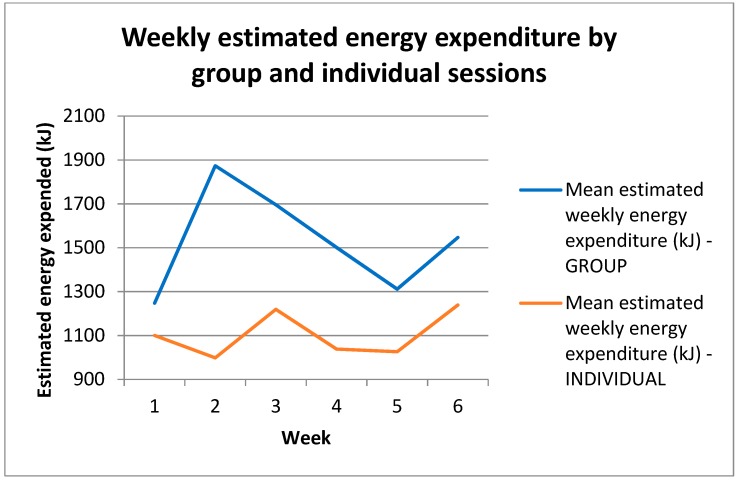
Weekly estimated energy expenditure by group and individual sessions. Estimated energy expenditure measured in kJ recorded during each PA session and averaged for each week of the intervention. Individual and group training days are shown separately. On average, estimated energy expenditure was significantly higher on group training days (*p* = 0.0016). kJ = kilojoules; *n* = 17.

**Figure 3 sports-07-00142-f003:**
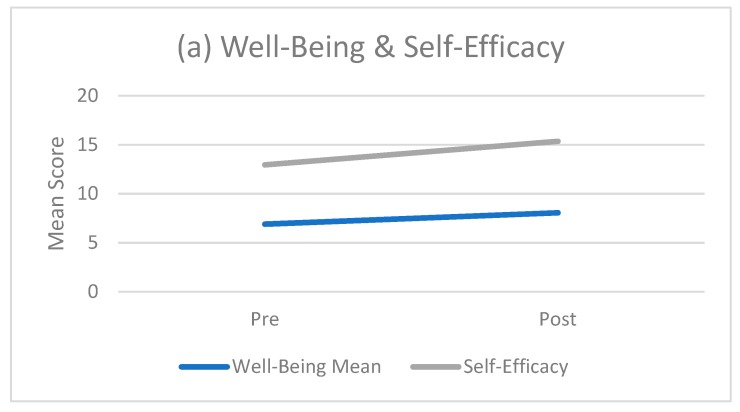
Changes in participant psychological scores pre- and post-GEx intervention. Chart (**a**) shows pre and post changes in well-being (using the Personal Wellbeing Index—Adult; *n* = 14) and self-efficacy (using The Physical Exercise Self-Efficacy Scale; *n* = 17) where improvements are represented by increased scores; chart (**b**) shows pre-post changes in depression, anxiety and stress (using the DASS21 questionnaire *n* = 17) where improvements are represented by decreased scores.

**Table 1 sports-07-00142-t001:** Itinerary for the 40-day GEx Program Commencing 30 April.

Week	Monday	Tuesday	Wednesday	Thursday	Friday	Saturday	Sunday
**Week 1** 800 kJ *		**WALK** Easy grade walk at River Torrens Linear Park; Includes ice-breakers	Individual	**GAITs** Group initiative challenges in Victoria Park	Individual	Individual	**KAYAK** Introductory Kayaking session at Garden Island
**Week 2** 1000 kJ *	Individual	**CYCLING** Introductory cycling session at linear park	Individual	**STRENGTH & STRETCH** Low impact strength exercises, stretching & yoga at Victoria Park	Individual	Individual	**CYCLING** Introductory trail riding at Brownhill Creek
**Week 3** 1200 kJ *	Individual	**WALK** Moderate grade walk at River Torrens Linear Park	Individual	**TEAM GAMES** soccer at Victoria Park	Individual	Individual	**WALK** Moderate walk at Marino Rocks
**Week 4** 1400 kJ *	Individual	**WALK** Moderate/challenging walk at River Torrens Linear Park	Individual	**CIRCUIT** Sweat track circuit in Victoria Park	Individual	Individual	**EXPLORING MORIALTA** Hike to the waterfalls; rock climbing and abseiling at Morialta CP
**Week 5** 1600 kJ *	Individual	**ORIENTEERING** Orienteering session in North Adelaide	Individual	**STRENGTH & STRETCH ** Strength, stretching & yoga at Victoria Park	Individual	Individual	**ORIENTEERING** Orienteering session at Belair National Park
**Week 6** 1800 kJ *	Individual	**WALK** Challenging walk on Torrens Linear Park	Individual	**GAITs** Group initiative challenges in Victoria Park	Individual	Individual	

* Target daily energy expenditure.

**Table 2 sports-07-00142-t002:** Participant information pre-intervention.

Variable	Mean	SD	Range
**Males (*n* = 8)**			
Age (years)	48.8	9.9	28.7–59.3
Height (cm)	178.2	5.0	167.5–183.4
Weight (kg)	93.3	16.9	71.0–120.5
Unweighted PA (min/week)	69	34	30–125
**Females (*n* = 9)**			
Age (years)	47.8	11.0	28.4–59.1
Height (cm)	161.3	4.4	153.2–168.4
Weight (kg)	80.5	17.7	60.7–117.6
Unweighted PA (min/week)	73	44	0–134
**All participants (*n* = 17)**			
Age (years)	48.3	10.2	28.4–59.3
Height (cm)	169.2	9.8	153.2–183.4
Weight (kg)	86.5	18.0	60.7–120.5
Unweighted PA (min/week)	71	38	0–134

Mean, standard deviation (SD) and range are shown.

**Table 3 sports-07-00142-t003:** Program of daily activities.

Week	Monday	Tuesday	Wednesday	Thursday	Friday	Saturday	Sunday
**1**	Pre-program	**WALK**	Individual Day	**INITIATIVE TASKS**	Individual Day	Individual Day	**KAYAK**
		HR119 1088 kJ	HR115 938 kJ	HR126 1410 kJ	HR123 1184 kJ	HR125 1135 kJ	HR112 1256 kJ
**2**	Individual Day	**CYCLING**	Individual Day	**STRENGTH & STRETCH**	Individual Day	Individual Day	**CYCLING**
	HR 120 1033 kJ	HR128 1649 kJ	HR138 1094 kJ	HR109 1949 kJ	HR121 915 kJ	HR118 875 kJ	HR128 2066 kJ
**3**	Individual Day	**WALK**	Individual Day	**TEAM GAMES**	Individual Day	Individual Day	**COASTAL WALK**
	HR 111 1049 kJ	HR119 1565 kJ	HR115 1213 kJ	HR119 1123 kJ	HR115 922 kJ	HR128 1513 kJ	HR132 2470 kJ
**4**	Individual Day	**WALK**	Individual Day	**CIRCUIT**	Individual Day	Individual Day	**EXPLORING MORIALTA**
	HR 121 1181 kJ	HR119 1296 kJ	HR131 874 kJ	HR134 1368 kJ	HR114 958 kJ	HR114 1100 kJ	HR109 1886 kJ
**5**	Individual Day	**ORIENTEERING**	Individual Day	**STRENGTH & STRETCH**	Individual Day	Individual Day	**ORIENTEERING**
	HR 121 1225 kJ	HR110 1299 kJ	HR114 792 kJ	HR103 1366 kJ	HR102 707 kJ	HR113 1129 kJ	HR124 1256 kJ
**6**	Individual Day	**WALK**	Individual Day	**INITIATIVE TASKS**	Individual Day	Individual Day	Post-program
	HR 122 1068 kJ	HR115 1334 kJ	HR122 991 kJ	HR122 1776 kJ	HR123 1501 kJ	HR124 1351 kJ	

Mean HR and estimated EE (shown in kJ) recorded for group and individual exercise sessions across the 40-day outdoor PA intervention. HR = heart rate; kJ = kilojoules. *n* = 17.

**Table 4 sports-07-00142-t004:** Significant changes in pre- and post-intervention measures (*p* < 0.05).

Variable	*n*	Pre Mean	Pre SD	Post Mean	Post SD	*p* (Paired *t*-Test)
**Anthropometric**						
Hip girth (cm)	17	110.2	14.3	109.2	14.3	0.036
**Cardio-Metabolic**						
Total cholesterol (mmol/L)	17	5.0	1.2	4.7	1.1	0.026
**Fitness**						
Aerobic fitness (mL·kg^−1^·min^−1^)	17	25.4	10.6	30.8	13.3	0.002
Dynamic stability ^#^	17	2.7	1.5	3.1	1.6	0.038
**Physical Activity**						
Moderate PA (min/week)	17	55	45	266	132	<0.001
Vigorous PA (min/week)	17	13	20	179	150	<0.001
Weighted PA (min/week)	17	84	43	624	367	<0.001
**Psychological ^#^**						
Well-being	14 ^ǂ^	6.9	2.1	8.1	2.4	<0.001
Depression	17	8.1	7.2	2.8	3.7	<0.001
Anxiety	17	4.4	3.3	2.5	3.0	0.042
Stress	17	10.6	5.8	6.0	3.9	0.004

Means and standard deviations (SD) are shown. ^#^ Determined by Wilcoxon Signed-Ranks test. ^ǂ^ Some results are not included as questionnaires were incomplete.

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
