# Peer review of "GOING GREEN: The Effectiveness of a 40-Day Green Exercise Intervention for Insufficiently Active Adults"

_sports, 2019, doi:10.3390/sports7060142_

Round 1
Reviewer 1 Report
p.p1 {margin: 0.0px 0.0px 5.0px 0.0px; line-height: 15.0px; font: 13.3px Arial; color: #000000; -webkit-text-stroke: #000000; background-color: #f4f4f4} p.p2 {margin: 0.0px 0.0px 5.0px 0.0px; line-height: 15.0px; font: 13.3px Arial; color: #000000; -webkit-text-stroke: #000000} span.s1 {font-kerning: none}
Glover N et al analyzed the effects of a 40-day green exercise intervention for insufficiently active adults. They showed that Green exercise (GEx) programs increase the physical activities of insufficiently active adults, and might be beneficial for their physical and mental health.
Major point
What was the primary outcome of this study? The authors described that the outcome of this study was the physical and mental measures (line 246, 278, 312). However, the authors also put adherence and compliance as the outcome. In general, only one primary outcome in one clinical trial should be set, and other outcomes should be described as secondary outcomes. The authors should describe which result was the primary outcome of this study.
If the primary outcome was the compliance, the authors should describe that the physical and mental measures (and other results regarding energy expenditure, heart rate, etc) were secondary outcomes. Furthermore, the authors should discuss the effects of different compliance on the physical and mental measures as the limitation.
If the primary outcome was the physical and mental scores, the authors could not make any conclusions because of the differences in compliance.
Minor point
The authors should describe their affiliation.
Author Response
Dear reviewer
Thank you for your feedback, and I do hope it has been attended to sufficiently. I have noted relevant changes based on your feedback below, and a revised manuscript has been uploaded with tracked changes relating to feedback from you and one other reviewer.
Nicholas and Scott.
Point 1: What was the primary outcome of this study?
Response: re-worked aim from line 82, "The primary aims of this study was to measure program adherence and exercise compliance among participants..."
Point 2: If the primary outcome was the compliance, the authors should describe that the physical and mental measures (and other results regarding energy expenditure, heart rate, etc) were secondary outcomes.
Response: re-worked aim from line 83, "Secondary aims included to determine the changes in a range of physical, physiological and psychological variables"
The opening to the discussion has also been amended to reflect these changes. See from line 283.
Point 3: Furthermore, the authors should discuss the effects of different compliance on the physical and mental measures as the limitation.
Response: re-worked aim from line 328, "A final limitation to acknowledge is that disparate compliance rates (between individuals, or by individuals from week-to-week or in group versus individual sessions) are likely to have resulted in varied impacts on the health and well-being outcomes recorded. Greater consistency in compliance rates among individuals and by individual participants across the program would allow for more confident conclusions to be drawn on the effectiveness of GEx to improve health and well-being."
Point 4: author afilliations have been added. "Affiliations: Nicholas Glover and Scott Polley are associated with the Alliance for Research in Exercise, Nutrition and Activity (ARENA) within the School of Health Sciences, University of South Australia"
Reviewer 2 Report
The authors have addressed the concerns raised in the initial review.
Author Response
Thank you.
Reviewer 3 Report
Thank you for addressing many of the suggestions of the reviewers. However, several comments were not addressed:
Lines 109-110: How estimations for energy expenditure were calculated is still not stated
Line 172: The following remains unaddressed--
"Pre-intervention PA was based on how many weeks? Was it just the week before the first test, or more?"
Also---Did the group exercise sessions alone equal 150 min/wk or more? I think this is important to discuss as the individual compliance with the outside exercise requirement was as low as 29% in week 5. Also, it may be best to discuss this very low individual compliance in the limitation section.
Author Response
Dear reviewer
Thank you for your feedback, and I do hope it has been attended to sufficiently. I have noted relevant changes based on your feedback below, and a revised manuscript has been uploaded with tracked changes relating to feedback from you and one other reviewer.
Nicholas and Scott.
Point 1: Lines 109-110: How estimations for energy expenditure were calculated is still not stated
Response: The methods section has been updated from line 159 including a reference to the Polar S610 manual. "Participants were instructed to program measured VO2max and HRmax values into the Polar S610 watch [50]. The watch uses this data and with its proprietary software estimates EE, accounting for subject gender. Crouter and colleagues [51] found that using actual measured values for VO2max and HRmax resulted in a 4% error (SD±10%) in EE."
Point 2:
Line 172: The following remains unaddressed--
"Pre-intervention PA was based on how many weeks? Was it just the week before the first test, or more?"
Response: methods update with a clearer statement that confirms it was just the one week: "Participants in this study met criteria for classification as insufficiently active (PA level<150min/week) by completing one Active Australia Survey [46], a 7-day recall questionnaire."
Point 3: Also---Did the group exercise sessions alone equal 150 min/wk or more? I
think this is important to discuss as the individual compliance with
the outside exercise requirement was as low as 29% in week 5.
Response line 299: "For this study, weekly compliance (≥150min/week) could be reached by attending the three group sessions only, which may have acted to demotivate participation in individual sessions where compliance was much lower, for example week five (29%)."
Also, it may be best to discuss this very low individual compliance in the limitation section.
Response line 328: "A final limitation to acknowledge is that disparate compliance rates (between individuals, or by individuals from week-to-week or in group versus individual sessions) are likely to have resulted in varied impacts on the health and well-being outcomes recorded. Greater consistency in compliance rates among individuals and by individual participants across the program would allow for more confident conclusions to be drawn on the effectiveness of GEx to improve health and well-being."
Round 2
Reviewer 1 Report
The authors answered my questions.
The author affiliations should be described in line 5-7.